# Early Cannon Development in Females of the “Sanmartinero” Creole Bovine Breed

**DOI:** 10.3390/ani14040527

**Published:** 2024-02-06

**Authors:** Arcesio Salamanca-Carreño, Pere M. Parés-Casanova, Mauricio Vélez-Terranova, Germán Martínez-Correal, David E. Rangel-Pachón

**Affiliations:** 1Facultad de Medicina Veterinaria y Zootecnia, Universidad Cooperativa de Colombia, Villavicencio 500001, Colombia; 2Institució Catalana d’Història Natural, 08001 Barcelona, Spain; 3Facultad de Ciencias Agropecuarias, Universidad Nacional de Colombia sede Palmira, Palmira 763531, Colombia; 4Asociación de Criadores de Bovinos de Razas Criollas y Colombianas de los Llanos Orientales, Villavicencio 500001, Colombia

**Keywords:** body measurements, native breed, morphological harmony, morphology, zoometry

## Abstract

**Simple Summary:**

In creole bovine breeds, it is essential to know the change in morphological traits related to the size of the animal; this morphological change is known as allometry. With allometric studies, the function of body structures and their relationship with the performance and survival of breeds and species can be analyzed. The aim of this study was to describe the allometric growth of certain body parts of the Sanmartinero creole bovine breed, from an early age. A total of 79 females with an age range of 0.5 to 10 years were studied. Body measurements were obtained individually: cannon perimeter, thoracic perimeter, body length, withers height, and body weight. Functional and production indices were obtained for animals aged more than 3 years: corporal index, anamorphosis index, dactylo thoracic index, and the relative thickness of the cannon. The correlation between body weight and cannon perimeter did not show any tendency towards a plateau. In other words, in the Sanmartinero bovine, a very early development of the cannon perimeter is detected, which could cause bone fragility in heavier animals. The data suggest a low harmony with appendicular bone development, that is, less robust shaft bones to support body mass. Therefore, the measurement of the thoracic perimeter, the withers height, and the thorax perimeter should be considered in genetic improvement programs of the Sanmartinero creole bovine.

**Abstract:**

The variation in the dimensions of the body of living beings in relation to their size, function, or shape is known as allometry. Allometry studies allow the analysis of the function of body structures and their relationship with the performance and survival of breeds and species. The aim of this study was to describe the ontogenetic characteristics of the weight of certain bone elements of the Sanmartinero creole bovine breed, from an early age (6 months) until maturity. A total of 79 females of the Sanmartinero creole bovine breed with an age range of 0.5 to 10 years were studied. Body measurements were obtained individually using standard procedures: cannon perimeter, thoracic perimeter, body length, withers height, and body weight. Functional and production indices were obtained for animals aged more than 3 years (*n* = 53): corporal index, anamorphosis index, dactylo thoracic index, and the relative thickness of the cannon. The correlation between body weight and the cannon perimeter showed no trend towards a plateau. In the Sanmartinero creole bovine, therefore, a very early development of the cannon perimeter is detected, which could lead to bone fragility in heavier animals. Therefore, data suggest little harmony with appendicular bone development, i.e., with less robust cannons which are those necessary to support body mass. It is suggested that in genetic improvement programs of the Sanmartinero creole bovine, the measurement of the cannon perimeter, the withers height, and the thoracic perimeter should be considered.

## 1. Introduction

The size of one body part is correlated with the size of other parts or the entire body using an exponential scale [1]. Allometry studies the changes in different traits associated with the variation in the body size of organisms [2,3]. That is, allometry shows the development (comparatively greater or lesser) of a structure with respect to other structure [4]. The evolution of allometry is fundamental for the generation of morphological diversity and evolutionary processes [1]. When growth is accompanied by changes in both proportion and size, it is known as a “relative growth pattern” or “allometric growth” (e.g., morphological changes during development) [2,3,5,6]. Allometry is based on multiplicative growth models for size measurements such as bone lengths and other linear distances, areas, weights, or organ volumes [7].

At least three types of allometry have been defined: static allometry, which is the result of the variation between individuals of the same population and age group [8]; ontogenetic allometry, which refers to the covariation between characters during the growth of the same individual [2]; and evolutionary allometry, which reflects the evolutionary covariation of different traits and species along the branches of the phylogeny [3].

Attempts to find mathematical models adjustable to growing variables are very numerous and have succeeded each other over time with greater or lesser success [9]. When an organism grows, two types of phenomena take place in it: (a) the increase in weight and volume over time, and (b) the modification of the proportions of the various morphological regions, organs, and systems, until the adult or stable state is reached [10]. But there is a difficulty in establishing a general model due to the complexity of the processes involved in growth and their variation at different levels: interspecific, population, and individual [11].

The modification of body proportions is the result of the uneven or relative growth of the different parts of the organism, and for its quantification, the most suitable model used is the allometric one [11]. This model is based on the principle that the relative growth rate between two variables remains constant. This makes it possible to relate both variables by means of a simple equation of potential type, which, by means of a logarithmic transformation, is converted into a line whose slope is the allometry coefficient [8,11]. It has been widely used in numerous disciplines: physiology, morphogenesis, evolution, etc. [8]. If the proportions of the morphometric traits must remain similar with increasing body size, the linear dimensions should scale with body mass to the power of 1/3 [1]. For a quadruped animal to be able to support its own weight, the diameter and length of its trunk must be proportional to the mass of its body raised to 0.37 and 0.25, respectively, considering an animal body as a cylinder (“elastic similarity”) [12].

The relationships between the increase of a dimension as a function of time can be expressed also by non-lineal growth curves [11]. As is well known, these curves have a sigmoid shape when the increase in dimension is manifested in absolute terms and is generally known as the logistic function [13]. Michaelis and Menten developed models of this type. The Michaelis–Menten option fits to the equation *y* = *ax*/(*b* + *x*) [14]. Also, it has been successfully applied to the growth of weight and linear dimensions in mammals. The Bertalanffy curves are not adequate, since in this model, the inflection point (1/3 of the asymptotic value) is already fixed [11].

The allometric relationships between morphometric traits and living mass have been investigated for several years in different animal species, both wild and domestic. In bovines, it was found that as the size of the animal increased, the proportion of bone decreased and that of muscle increased [15]. In *Bos primigenius taurus* and *Bos primigenius indicus* cows, a positive allometry of the body mass ratio between males and females was demonstrated [16].

A comparative analysis of allometry between body weight and chest circumference between different species of animals has explained that the allometric slope is invariant between species [1]. In lambs, a positive allometry of the weight of different body parts with the axial skeleton in the postnatal period and a negative allometry regarding the skull and limbs in the prenatal period were revealed [11]. A study conducted on domestic chicken breeds (*Gallus gallus domesticus*) did not show allometry between size and body mass [17], while other studies on several Galliformes species found allometry for beak size and skull size [18]. Although allometry between morphological characters is an important source of diversification [19], and for the study of evolution and development, the adaptive significance of the change in the proportions between them is still being discussed [3,20,21].

On the other hand, morphometric measurements allow the analysis of the relationship between individuals and the evaluation of the harmony of the morpho structural model [22,23]. With morphometric measurements, zoometric indices (relationship between two variables) are constructed that are used to determine the zootechnical purposes of the animal [22,24,25]. Zoometric indices provide ethnological and productive information about animals [22,26], obtaining a basis for the classification of harmonic types within the breed [24,27,28]. Animal genetic resources, due to their diversity, contribute to human needs in the supply of food and raw materials, and constitute a genetic heritage for a country [29,30].

In the department of Meta, Colombian Orinoquia, there is the Sanmartinero creole bovine breed, which was formed from the bovines introduced by the Spanish in the 15th century [31]. Since its introduction to Latin America, it has experienced a prolonged process of natural selection that has allowed it to develop adaptive characteristics (e.g., reproductive efficiency) to live in extreme climatic environments (high temperature and relative humidity) and to feed on fibrous forages in the ecosystem where it lives [31,32,33].

The Sanmartinero creole bovine has qualities such as docility and the ability to travel long distances and to defend itself from attacks by predatory animals [31]. It produces high quality meat, with tenderness being its main characteristic, while the milk contains Kapa casein ß, a higher amount of total solids, fats, and protein [34]. The Sanmartinero creole bovine, due to its adaptive condition, represents the best competitive and sustainable alternative for producers to improve the production and profitability of farms. It is considered a biological and economic heritage for the efficient provision of food (meat and milk), skin, and work [33].

For the Sanmartinero creole bovine breed, research has focused on genetic traits of growth [32], performances and productivity [34], as well as some biometric data [31,35]. However, to date and to the authors’ knowledge, there are no published works on their allometric changes, so this work has a specific interest in the breed, as well as a general interest in allometry in bovine breeds. As the allometric study is essential to understand developmental processes, the aim of this work is to describe the ontogenetic characteristics of the weight of certain bone elements of this breed, from an early age (6 months) until maturity.

## 2. Material and Methods

### 2.1. Study Region

This study was conducted in the department of Meta in Colombia (part of the Orinoquia region) (altitude: 01°36′29″ and 04°54′24″ N; longitude 71°04′42″ and 74°54′09″ W; altitude from 200 to 450 m). The region belongs to humid tropical and very humid tropical forest zones. Its topography is flat and undulating, featuring acidic soils, with mineral deficiencies, especially P, Cu, and Zn, and high Al contents. The average environmental temperature is 26 °C, with a relative humidity of 87% in the rainy period (April to November) and 55% in the dry period (November to March). Rainfall varies between 2.700 mm (highlands) and 3.500 mm (foothills of the plains) [33]. The most important economic sector of the department is extensive livestock farming.

### 2.2. Data Source

A total of 79 females of the Sanmartinero creole bovine breed with an age range of 0.5 to 10 years were studied. The animals came from three different farms located in the department of Meta, Colombian Orinoquia. Males were not used because livestock farmers sell them at weaning time, so it was difficult to find them, at least in the farms visited.

The farms where the animals were measured belong to livestock farmers associated with the Association of Creole and Colombian Cattle Breeders of the Eastern Plains (ASOCRIOLLANOS, for its acronym in Spanish). The farms were chosen out of convenience after a meeting with the livestock farmer association. Direct mating with several bulls is used on all farms. The main activity is breeding; however, on some farms milking is carried out by hand with the calf present.

A peculiarity of economic importance in the Sanmartinero creole females is their ease of calving since the expulsion of the fetus, with the cow in a standing position in 86 percent of the cases, occurs in about 5 min, which is related to the small size of calves at birth. The breed has the capacity to live in large areas, where forage is scarce and waters are distant, and has a certain tolerance to ectoparasites [31]. Grazed native grasses are the main food source for the animals: hairy grass (*Trachypogon vestitus*), guaratara grass (*Axonopus purpussi*), lambedora grass (*Leersia hexandra*), black grass (*Paspalum plicatulum*), grass Carretera (*Paratheria prostrata*), gramalote grass (*Paspalum fasciculatum*), and introduced grasses *Urochloa* spp., among others. Its diet is also complemented with mineralized salt.

### 2.3. Morphometric Measures

For each animal, 5 linear variables were obtained: the metacarpal perimeter or cannon perimeter (CaP) and the thoracic perimeter (TP) with a measuring tape, the body length (BL) and the heigh at the withers (WH) with a zoometric cannon (Figure 1), and the body weight (BW) with a scale, using standard procedures [36].

Cannon perimeter (CaP: perimeter in the narrowest part of the metacarpal bones III and IV fused in its middle and proximal thirds);Thoracic perimeter (TP: measurement of the girth around the chest, passing through the withers and the sternum);Body length (BL: distance between the most cranial and lateral point of the humeral joint and the most caudal ilio-ischial point);Withers height (WH: distance from the floor to the highest point of the withers-interscapular region, 3rd, and 4th spinous process of the thoracic vertebrae);All measurements were taken by two students who participated as research assistants.

### 2.4. Statistical Analysis

Bivariate correlations were performed with a r_s_ Spearman correlation. Linear regressions were established as *Y* = b + *X*^a^ with an ordinary least square (LS). Finally, a set of 2 indices related to the productive capacity of meat or milk (dactylo thoracic index and the relative thickness of the cannon) and 2 ethnological indices (corporal index and anamorphosis index) [24,37]. The indices were obtained for females aged more than 3 years (*n* = 53), using the following formulas:Corporal Index (CI) = body length/thoracic perimeter × 100
Anamorphosis Index (AI) = (thoracic perimeter)^2^/withers height
Dactylo Thoracic Index (DTI) = cannon perimeter/thoracic perimeter × 100
Relative Thickness of the Cannon (RTI) = cannon perimeter/withers height × 100.

Data were analyzed with the program PAST v. 2.17c [38]. A confidence level of 95% was used in all cases.

## 3. Results

The body weight ranged from 100 to 566 kg (X = 363.5 ± 103.4 kg), while the cannon perimeter ranged between 15 and 38 cm (X = 26.7 ± 4.17 cm), which did reflect a bivariate correlation (r_s_ = 0.355; *p* = 0.0012). The cannon perimeter also appeared to be correlated with age (r_s_ = 0.275; *p* = 0.0139).

As stated in the introduction section, the growth of a given magnitude in relation to weight (cannon perimeter to body weight, in our case) follows a sigmoid-like curve, although not clearly shown in the Michaelis–Menten model (Figure 2).

The linear regression of cannon perimeter to body weight showed an exponent of much less than 1/3 (*Y* = 23.7 + *X*^0.008^, for *Y* = cannon perimeter, and *X* = body weight), which was not significant (*r* = 0.204; *p* = 0.068) (Figure 3). BW was highly correlated with age (r_s_ = 0.725; *p <<* 0.001).

The zoometric indices (Table 1) show a high variability in a range from 8.7% for the corporal index to 18.7% for the relative thickness of the cannon. The anamorphosis index presented less variability (13.9%). In general, the four indices reflected a low phenotypic homogeneity among the animals evaluated.

The anamorphosis index determines the conformation of the individual. The value obtained in this study is considered low (2.53%; <30%).

The corporal index refers to the body length of cattle (dairy, more elliptical; beef, more circular). The value shown in this study (84.11%) was slightly higher than that reported for dairy cattle (78–83%; see discussion).

The dactylo thoracic index was very high (15.68%). When combining this index with the relative thickness of the cannon index (22.90%), they suggest little harmony with bone development.

## 4. Discussion

The perimeter of the cannon is related to the silhouette of the animal, and those animals with straight profiles tend to have intermediate cannon perimeters [27]. On the other hand, the perimeter of the cannon has a differentiating value between breeds destined for milk and meat production [36]. The breeds for milk production usually have medium-thick or really fine cannons, while in breeds for meat production the trend is from medium to large [36,39]. According to the previous criteria, the Sanmartinero creole bovine could occupy an intermediate position, that is, with dual meat–milk aptitude.

The cannon perimeter is important because it supports the weight of the animal. For this bone, allometric coefficients reveal that there is the same cannon thickness independently of age and body weight. That is to say, from the point of view of the skeleton, after 6 months of age, the development of cannon is proportionally reduced. So, the data suggest that although a good structure for milk production appears in the Sanmartinero cow, there is little limb harmony in the cannon development, with a tendency to “close the metacarpal bone” too early in the animal’s life. But it should not be inferred from this that an increase in the volume of the limbs, an “excess of bone”, is always desirable, since the quality and shape of the bones [22,36], as well as of the joints and tendons, must also be considered.

The corporal index refers to the body length and the circumference of the thorax, determining the shape of the thoracic section according to the type of cattle, dairy (more elliptical) and meat (more circular), although they are completely different in their numerical values [24,28]. The corporal index (84.11%) was slightly higher than the range reported for dairy cattle (78–83%) and much higher for the meat type (64–70%) [28]. Thus, the Sanmartinero creole bovine seems to be in a position of dual aptitude, as mentioned above. The corporal index was higher than that reported for the Limonero creole bovine from Venezuela (74.3%) [24], the Mixtec creole bovine from Oaxaca (79.9%) [25], the creole bovine Guaymi of Panama (80.52%) [40], the Barroso-Salmeco creole of Guatemala (83.55%) [41], the Casanare creole bovine from Colombia (82.5%) [42], the Saavedra creole bovine from Bolivia (82%) [43], and the creole bovine from the Andean highlands of Peru (82.98%) [44]. However, the corporal index was lower than that reported in Ecuador for the creole bovine from the Province of Azuay (86.8%) [45], the creole bovine from the Province of Huamanga (88.97%) [37], the creole bovine of the Ecuadorian coast (87.5) [46], the creole bovine from Uruguay (88.2%) [47], the creole bovine from the Province of Santa Elena, Ecuador (95.03%) [48], and the creole bovine from the Province of Loja (115.9%) [26].

In bovine, when the value of the anamorphosis index is high (4.0 to 5.0), it is considered to be a meat animal, while if it is lower (2.5 and 3.0), the animal tends to be a milk animal [22,24,25]. As the anamorphosis index decreases, the animal is considered tall of its legs and with a lower body weight [22]. In our study, the anamorphosis index found (2.53%) corresponds to a longer animal and was located within the same range of values obtained in other Ibero-American creole bovine: the Limonero creole bovine from Venezuela (2.5%) [24], the Saavedra creole bovine (2.7%) [43], the Mixtec creole bovine (2.1%) [25], the creole bovine from the province of Azuay, Ecuador (2.2%) [45], the creole bovine from the Andean highlands, Peru (2.01%) [44], the Casanare creole bovine (2.03%) [42], and the creole bovine from the Province of Huamanga (1.86%) [37]. Therefore, the Sanmartinero creole bovines phenotypically tend to be more milk producers than meat producers, with an anamorphosis index similar to the Limonero creole bovine of Venezuela.

According to the indices related to productive aptitudes, the dactylo thoracic index provides an idea of the degree of fineness of the skeleton, with its value being high in dairy animals (11–12%) and low in meat-type animals (less 10%) [24,27]. The dactylo thoracic index relates the strength of the limbs to the body mass they support [27,28]. In the case of the Sanmartinero creole bovine, the dactylo thoracic index (15.68%) was higher compared to that reported for other Ibero-American creole bovine: the creole bovine of the Ecuadorian coast (11.3%) [46], the Mixtec creole bovine of Mexico (11.0 %) [25], the Barroso-Salmeco creole bovine of Guatemala (10.95%) [41], the creole bovine from the Province of Huamanga (10.78%) [37], the creole bovine from the Andean highlands, Peru (10.49%) [44], the creole bovine Guaymi of Panama (10.32%) [40], the Limonero creole bovine from Venezuela (10.14%) [24], the creole bovine of the province of Azuay, Ecuador (10.3%) [45], the Saavedra creole bovine (10%) [43], and the creole bovine from the province of Santa Elena, Ecuador (9.8%) [49].

On the other hand, the study of the morphometry of structures in animals provides a solid basis to establish allometric relationships between structure and body size, helping to understand the laws of growth [50]. In addition, it allows researchers to know the morphology that the animal has acquired over time and its productive capacities [47]. The external morphology has value as a descriptor, identifier, and differentiator of an individual or breed and allows a greater productive assessment of the animal [51]. On the other hand, morphometric characteristics in creole bovine provide sufficient technical information in the short term [44]. However, at the level of Ibero-America, the lack of information on creole bovine is based both on characterization studies and on the productive behavior of these animals [52].

All over the world there are genetically related but morphologically distinct breeds that may differ in size and shape [16]. Allometric studies have ecological and evolutionary implications [50]. The evolution of allometry is essential for the generation of morphological diversity [1]. It is important to highlight that allometric studies allow the evaluation of the development of different morphological structures throughout the life of an individual [4]. The development of morphological structures can be affected by nutritional level, age, sex, and historically by directed selection and domestication [51].

In the present study, an allometry of the perimeter of the cannon with body weight has been evidenced, a characteristic that was unknown for the Colombian Sanmartinero creole bovine. The results on the behavior of the cannon perimeter with age would indicate that, in a future genetic improvement program for this creole bovine breed, special attention should be paid to the metacarpal perimeter, since as age advances, if there is no development proportionally, animals with a weak cannon could present difficulties in movement and supporting their body weight. Therefore, the result is substantial and encourages us to continue to research joint variations and the functional, evolutionary, and adaptive characteristics of Sanmartinero creole bovine. Likewise, the result serves as an example for future studies in other breeds of Colombian creole bovine. Studies should continue in males and in another regions of the country. This result is of interest for farmers who should consider the perimeter of the cannon when selecting their animals.

## 5. Conclusions

In this study, the Sanmartinero creole cow shows a good structure for milk production, but there is little harmony of the limbs in the development of the cannon. Although the results reflected allometry, a phenomenon that had not been reported for the breed, its possible causes are still unknown. Finally, the measurement of the perimeter of the cannon, the height at the withers, and the perimeter of the thorax must be considered in genetic improvement programs of the Sanmartinero creole bovine.

## Figures and Tables

**Figure 1 animals-14-00527-f001:**
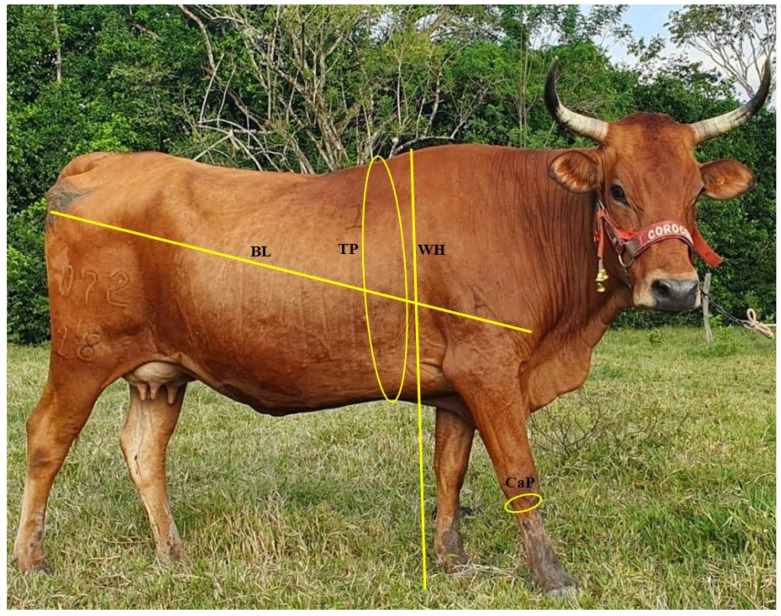
Anatomical points where the corporal measurements for the present study were taken. CaP = Cannon perimeter; TP = Thoracic perimeter; BL = Body length; WH = Withers height. Photograph taken at the Punta Hermosa farm. Source: authors.

**Figure 2 animals-14-00527-f002:**
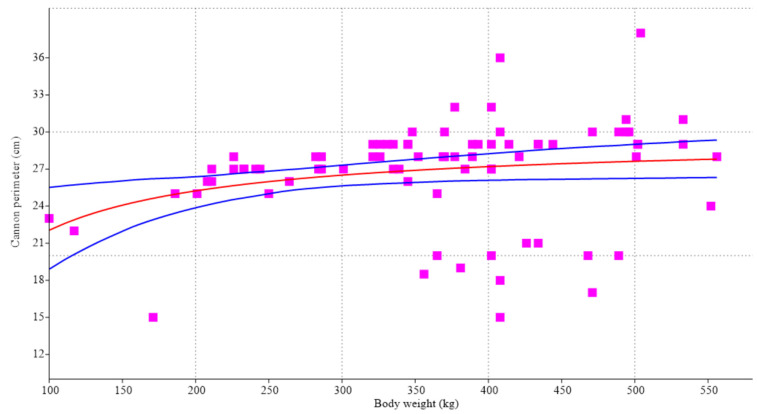
Model based on the Michaelis–Menten equation (red line ± 95% confidence levels in blue) for 79 females (
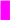
) of the Sanmartinero creole bovine breed, using body weight and cannon perimeter as variables. The model clearly appears as a rather constant plateau independent of weight gain.

**Figure 3 animals-14-00527-f003:**
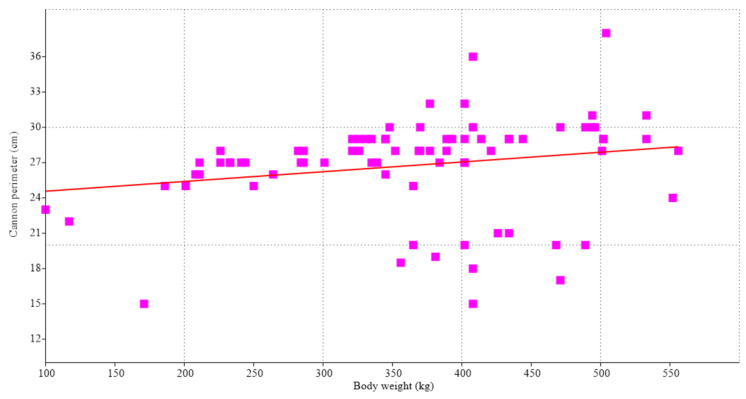
Linear regression (red line) of cannon perimeter to body weight for 79 females (
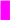
) of the Sanmartinero creole bovine breed, which showed an exponent of much less than 1/3 (*Y* = 23.7 + *X*^0.008^, for *Y* = cannon perimeter, and *X* = body weight) and was not significant (*r* = 0.204; *p* = 0.068).

**Table 1 animals-14-00527-t001:** Values for indices for 53 Sanmartinero creole bovine breed females aged more than 3 years. See text for formulas.

Index	Average (%)	CV (%)
Corporal Index (CI)	84.11	8.7
Anamorphosis Index (AI)	2.53	13.9
Dactylo Thoracic Index (DTI)	15.68	18.6
Relative Thickness of the Cannon (RTI)	22.90	18.7

CV = Coefficient of variation.

## Data Availability

Data are available upon reasonable request to the second author.

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
