# Peer review of "Early Cannon Development in Females of the “Sanmartinero” Creole Bovine Breed"

_animals, 2024, doi:10.3390/ani14040527_

Round 1

Reviewer 1 Report

Comments and Suggestions for Authors

The manuscript is a great contribution to the knowledge of zoometry in a Colombian bovine breed. Besides its application to genetic improvement programs of the Sanmartinian breed, essential to this region of Colombia. I have some revisions to this manuscript as below:

Line 96: Are the correct names Bos taurus and Bos indicus? I think that is Bos taurus taurus and Bos taurus indicus, respectively.

Line 102: Change the term “extremities” to “limbs”

Line 102-105: Why do you include allometry studies in birds if this study is in a breed of a mammal species? I think that these references should not be here or include other examples of allometry performed in mammals.

Line 36: I recommend changing the redaction to avoid including the Colombia map in the figure 1. e.g.: …department of Meta in Colombia (Part of the Orinoquia region)...

Lines 144-147: I think that this figure is not necessary. The future readers of this manuscript can search the Colombia map to find the department of Meta, even the regions of Colombia.

Lines 155-167: This paragraph should be in the introduction. Besides, it should not invite to review the web page but cite it. Please cite it and include it in references based on the author guidelines of Animals-MDPI

Line 168: This figure is not necessary either. It would be more interesting to include photographs including the anatomical points where were taken the measurements for the present study (lines 186-193).

Lone 186: … metacarpal bone (metacarpal bones III and IV fused).

Line 187: change the term “upper” to “proximal”.

Line 211: .Cannon… (upper letter to "C")

Lines 249-251: Please include references to support this information.

Lines 278-282: Include the reported indexes of those authors for each creole bovine. It would be interesting to perform a statistical analysis to review if the corporal indexes have actually significant differences among creole breeds. This will help to support this paragraph and the manuscript.

Line 286: Our study… (correct ours)

Author Response

First reviewer’s responses

Dear reviewer

The authors appreciate the insightful comments.

We attach the corrections and responses and were inserted into the text.

Comment

The manuscript is a great contribution to the knowledge of zoometry in a Colombian bovine breed. Besides its application to genetic improvement programs of the Sanmartinian breed, essential to this region of Colombia. I have some revisions to this manuscript as below:

Line 96: Are the correct names Bos taurus and Bos indicus? I think that is Bos taurus taurus and Bos taurus indicus, respectively.

Response. It is correct Bos taurus and Bos indicus

Line 102: Change the term “extremities” to “limbs”

 Response. Corrected in the text.

Line 102-105: Why do you include allometry studies in birds if this study is in a breed of a mammal species? I think that these references should not be here or include other examples of allometry performed in mammals.

Response. OK; but we place the references as examples of studies of allometries found in other species. We consider it to be interesting for many readers, farmers, and animal producers.

Line 36: I recommend changing the redaction to avoid including the Colombia map in the figure 1. e.g.: …department of Meta in Colombia (Part of the Orinoquia region).

Response. Corrected in the text.

Lines 144-147: I think that this figure is not necessary. The future readers of this manuscript can search the Colombia map to find the department of Meta, even the regions of Colombia.

 Response. The figure was removed.

Lines 155-167: This paragraph should be in the introduction. Besides, it should not invite to review the web page but cite it. Please cite it and include it in references based on the author guidelines of Animals-MDPI

Response. Corrected, paragraph was changed to Introduction.

Line 168: This figure is not necessary either. It would be more interesting to include photographs including the anatomical points where were taken the measurements for the present study (lines 186-193).

 Response. The figure was changed. A photograph was added including the anatomical points where the measurements were taken.

Lone 186: … metacarpal bone (metacarpal bones III and IV fused).

Response. Corrected in the text.

Line 187: change the term “upper” to “proximal”.

 Response. Corrected in the text.

Line 211: .Cannon… (upper letter to "C")

Response. Corrected in the text.

Lines 249-251: Please include references to support this information.

Response. The information was removed according to the review's recommendations.

Lines 278-282: Include the reported indexes of those authors for each creole bovine. It would be interesting to perform a statistical analysis to review if the corporal indexes have actually significant differences among creole breeds. This will help to support this paragraph and the manuscript.

Response. The indices reported for each Creole bovine were added. The information was included in the text. In future studies, statistical analysis will be carried out to check whether the body indices show significant differences between Creole cattle. Thank you for your recommendation

Line 286: Our study… (correct ours)

Response. Corrected in the text.

Reviewer 2 Report

Comments and Suggestions for Authors

The article, despite systematically addressing regional data regarding a breed of cattle, does not address a scientific gap. The authors extrapolate the data obtained in a specific and punctual way. In this way, it does not elucidate the general data necessary to clarify the cattle in question.

Author Response

Second reviewer’s responses

Dear reviewer

The authors appreciate the insightful comments.

We attach the corrections and responses and were inserted into the text.

 Comment

The article, despite systematically addressing regional data regarding a breed of cattle, does not address a scientific gap. The authors extrapolate the data obtained in a specific and punctual way. In this way, it does not elucidate the general data necessary to clarify the cattle in question.

“the authors extrapolate the data to a general statement. revealing data fragility. mainly due to the possibility of other factors that must be evaluated”

Response. Dear reviewer, we are supporting ourselves in accordance with what has been reported for other Ibero-American creole bovine.

In the text it says:

“Our study the anamorphosis index found (2.53 %) corresponds to a longer animal and was located within the same range of values obtained in other Ibero-American creole bovine: Limonero creole from Venezuela (2.5 %) [Ref. 24], Saavedra creole bovine (2.7 %) [Ref. 43], Mixtec creole bovine (2.1 %) [Ref. 25], creole from the province of Azuay, Ecuador (2.2 %) [Ref. 45], creole bovine from the Andean highlands, Peru (2.01 %) [Ref. 44], and Casanare creole bovine (2.03 %) [Ref. 42]. Therefore, the Sanmartinero creole bovine phenotypically tends to be more producers of milk than meat”.

In the ABSTRACT, what was crossed out was eliminated:

“This study encourages us to continue investigating the joint variations of the functional, 45 evolutionary, and adaptive characteristics of this creole bovine”.

Comment

“clarify the items evaluated, present in the body of the figure or in the caption”

Response. Dear reviewer, in the title it´s written body weight and cannon perimeter

In the RESULTS, what was crossed out was eliminated:

“... showing an animal with a predisposition to produce more milk than meat”.

“This indicates that the Sanmartinero creole bovine can be considered an animal with a tendency to produce more milk than meat. In a tropical livestock production system, it could be considered a dual-purpose animal (milk and meat)”.

“is, with cannon less robust than those necessary to support body mass”.

In the DISCUSSION, what was crossed out was eliminated:

“In the period of formation of the calf and therefore of the skeleton, there are different moments of growth. There is a stage in which the organs grow proportionally to the rest of the body and another in which they develop more rapidly. Isometric growth refers to the stage in which the tissue grows at the same rate as the animal's live weight, which in the case of the cannon perimeter is up to the first 3 months of life”.

“This fact could functionally lead to bone fragility in females as they gain weight with age”.

In the CONCLUSIONS, what was crossed out was eliminated:

“This particularity could functionally lead to bone fragility in Sanmartinero cows as they gain weight with age”.

Reviewer 3 Report

Comments and Suggestions for Authors

Sanmartinero cattle are known for their outstanding quality and productivity. It originates from the Orinoquia region of Colombia and developed through natural selection and effective zootechnical methods used by the Jesuits back in the seventeenth century. These cattle were formed by crossing Spanish cows introduced in the fifteenth century with local species. Sanmartinero is distinguished by its outstanding qualities and productivity. It is considered a purebred breed and performs excellently when crossed with other breeds, particularly commercial Cebu C type Brahman. Research has shown that crossing with Sanmartineros results in a heterosis effect, meaning that the offspring have better characteristics compared to the parent breeds. Sanmartinero is also known for its adaptability to various conditions such as diseases, parasites and climatic conditions. It is highly resistant and can be preserved for a long time, and also has high fertility. Moreover, its ability to cross with other breeds allows it to produce offspring with improved characteristics. Based on all the studies carried out and the positive results, it can be concluded that Sanmartinero is the most competitive and sustainable alternative to improve productivity, profitability and quality of meat, as well as for the development of breeding and dairy farming in the Orinoquia region of Colombia.

The authors have done a lot of scientific work. A total of 79 females with an age range of 0.5 to 10 years were studied. This is the correct age range.

There are some comments that need to be corrected. After this, the work can be accepted for publication.

Row 34 - In the abstract, the authors indicate the “mean age” of the cows. Thus the reader is confused. It is not gut. Such information does not give anything. This is the same if we write that 79 cows have an “mean temperature” of 38.5 (for 40 cows it is 38, for the rest it is 41.5). This is not correct.

Row 98 - «A comparative analysis of the allometry of body weight and chest circumference between cattle breeds and species showed that the allometric slope is invariant across species [1].» - This sentence talks about different breeds and species of cattle.......and at the end just about the species. Here we need to add... «different species of animals». And the next sentence starts about lambs.

Row 150 – «A total of 79 females with a mean age of 4.1 ± 2.2 years (age range 0.5 to 10 years) of the Sanmartinero creole bovine were studied.». Needs to be removed – « with a mean age of 4.1 ± 2.2 years».

Author Response

Third reviewer’s responses

Dear reviewer

The authors appreciate the insightful comments.

We attach the corrections and responses and were inserted into the text.

 Comment

Sanmartinero cattle are known for their outstanding quality and productivity. It originates from the Orinoquia region of Colombia and developed through natural selection and effective zootechnical methods used by the Jesuits back in the seventeenth century. These cattle were formed by crossing Spanish cows introduced in the fifteenth century with local species. Sanmartinero is distinguished by its outstanding qualities and productivity. It is considered a purebred breed and performs excellently when crossed with other breeds, particularly commercial Cebu C type Brahman. Research has shown that crossing with Sanmartineros results in a heterosis effect, meaning that the offspring have better characteristics compared to the parent breeds. Sanmartinero is also known for its adaptability to various conditions such as diseases, parasites and climatic conditions. It is highly resistant and can be preserved for a long time, and also has high fertility. Moreover, its ability to cross with other breeds allows it to produce offspring with improved characteristics. Based on all the studies carried out and the positive results, it can be concluded that Sanmartinero is the most competitive and sustainable alternative to improve productivity, profitability and quality of meat, as well as for the development of breeding and dairy farming in the Orinoquia region of Colombia.

The authors have done a lot of scientific work. A total of 79 females with an age range of 0.5 to 10 years were studied. This is the correct age range.

There are some comments that need to be corrected. After this, the work can be accepted for publication.

Row 34 - In the abstract, the authors indicate the “mean age” of the cows. Thus the reader is confused. It is not gut. Such information does not give anything. This is the same if we write that 79 cows have an “mean temperature” of 38.5 (for 40 cows it is 38, for the rest it is 41.5). This is not correct.

Response. Corrected in the text. It´s written: ...with an age range of 0.5 to 10 years were studied.

Row 98 - «A comparative analysis of the allometry of body weight and chest circumference between cattle breeds and species showed that the allometric slope is invariant across species [1].» - This sentence talks about different breeds and species of cattle.......and at the end just about the species. Here we need to add... «different species of animals». And the next sentence starts about lambs.

Response. Corrected in the text.

Row 150 – «A total of 79 females with a mean age of 4.1 ± 2.2 years (age range 0.5 to 10 years) of the Sanmartinero creole bovine were studied.». Needs to be removed – « with a mean age of 4.1 ± 2.2 years».

Response. Corrected in the text. It´s written: ...with an age range of 0.5 to 10 years were studied.

Round 2

Reviewer 1 Report

Comments and Suggestions for Authors

My suggestions were performed. I only have some minor revisons:

Line 328:... in another region..

Line 94: Review the below article, which was based on mitocondrial ADN, the cattle lines are suggested as two subspecies: "These data suggest domestications from several differentiated populations of B. primigenius and a subspecies status for taurine (B. primigenius taurus) and zebuine (B. primigenius indicus) cattle. " by Hiendleder et al.

Hiendleder S, Lewalski H, Janke A. Complete mitochondrial genomes of Bos taurus and Bos indicus provide new insights into intra-species variation, taxonomy and domestication. Cytogenet Genome Res. 2008;120(1-2):150-6. doi: 10.1159/000118756. Epub 2008 Apr 30. PMID: 18467841 .

According to said article the names are Bos primigenius indicus and Bos primigenius taurus. 

Author Response

First reviewer’s responses (Round2)

Dear reviewer

The authors appreciate the insightful comments.

We attach the corrections and responses and were inserted into the text.

Comment

My suggestions were performed. I only have some minor revisons:

Line 328:... in another region..

Response. Corrected in the text.

Line 94: Review the below article, which was based on mitocondrial ADN, the cattle lines are suggested as two subspecies: "These data suggest domestications from several differentiated populations of B. primigenius and a subspecies status for taurine (B. primigenius taurus) and zebuine (B. primigenius indicus) cattle. " by Hiendleder et al.

Hiendleder S, Lewalski H, Janke A. Complete mitochondrial genomes of Bos taurus and Bos indicus provide new insights into intra-species variation, taxonomy and domestication. Cytogenet Genome Res. 2008;120(1-2):150-6. doi: 10.1159/000118756. Epub 2008 Apr 30. PMID: 18467841 .

According to said article the names are Bos primigenius indicus and Bos primigenius taurus. 

Response. Corrected in the text Bos primigenius taurus and Bos primigenius indicus.

Reviewer 2 Report

Comments and Suggestions for Authors

The article "Early cannon development in females of the creole 'Sanmartinero' bovine breed" has been corrected punctually. However, it exhibits data extrapolation in a general sense. Despite the modifications, it is still not suitable for publication in the journal.

Author Response

Second reviewer’s responses (Round2)

Dear reviewer

The authors appreciate the insightful comments.

We attach the corrections and responses and were inserted into the text.

 Comment

The article "Early cannon development in females of the creole 'Sanmartinero' bovine breed" has been corrected punctually. However, it exhibits data extrapolation in a general sense. Despite the modifications, it is still not suitable for publication in the journal.

Response. The text was improved in the discussion.
